**Understanding nitrate formation in a world with less sulfate.**

Petros Vasilakos[1], Armistead Russell[2], Rodney Weber[3], and Athanasios Nenes[1,3,4,5†]

[1] School of Chemical and Biomolecular Engineering, Georgia Institute of Technology, Atlanta, Georgia, 30332, USA

[2] School of Civil and Environmental Engineering, Georgia Institute of Technology, Atlanta, Georgia, 30332, USA

[3] School of Earth and Atmospheric Sciences, Georgia Institute of Technology, Atlanta, Georgia, 30332, USA

[4] Institute of Chemical Engineering Sciences, Foundation for Research and Technology-Hellas, Patras, GR 26504, Greece

[5] Institute for Environmental Research and Sustainable Development, National Observatory of Athens, Palea Penteli, GR 15236, Greece

[†] Corresponding Author: A. Nenes (athanasios.nenes@gatech.edu)

**Abstract**

3        $SO_2$ emission controls, combined with modestly increasing ammonia, have been thought

to generate aerosol of significantly reduced acidity where sulfate is partially substituted by nitrate.
However, neither expectation agrees with decadal observations in the Southeastern US, suggesting
that a fundamentally different response of aerosol pH to emissions changes is occurring. We
postulate this "nitrate substitution paradox" arises from a positive bias in aerosol pH in model
simulations. This bias can elevate pH to where nitrate partitioning is readily promoted, leading to
behavior consistent with "nitrate substitution". CMAQ simulations are used to investigate this
hypothesis; modeled PM2.5 pH using 2001 emissions compare favorably with pH inferred from
observed species concentrations. Using 2011 emissions, however, leads to simulated pH increases
of one unit, which is inconsistent with observations from that year. Non-volatile cations ($K^+$, $Na^+$,
$Ca^{+2}$, and $Mg^{+2}$) in the fine mode are found responsible for the erroneous predicted increase in
aerosol pH of about 1 unit on average over the US. Such an increase can induce a nitrate bias of
1-2 $\mu g\ m^{-3}$ which may further increase in future projections, reaffirming an otherwise incorrect
expectation of a significant "nitrate substitution". Evaluation of predicted aerosol pH against
thermodynamic analysis of observations is therefore a critically important, but overlooked, aspect
of model evaluation for robust emissions policy.

## Introduction

Aerosol acidity is a driver of many important atmospheric processes (Guo et al. 2015, Weber et al. 2016), catalyzing the conversion of isoprene oxidation products to form secondary organic aerosol (SOA) (Xu et al. 2015, Pye et al. 2013, Surratt et al. 2010, Eddingsaas et al. 2010) and driving the semi-volatile partitioning of key aerosol species (Guo et al. 2015, Weber et al. 2016). High acidity can also lead to the solubilization of iron, copper and other trace metals in aerosol which may serve as nutrients for ecosystems (Meskhidze et al. 2003), but also prove toxic for humans (Ghio et al. 2012, Fang et al. 2017). Significant reductions in primary pollutant emissions over the last decades has greatly improved air quality in the developed world, and is also thought to fundamentally affect aerosol acidity. $SO_2$, an important aerosol precursor and a major driver of its acidity, has seen decreases of about 6% $yr^{-1}$ over the 2001-2011 period alone in the US, with a continued anticipated downward trend (Pinder et al. 2007, 2008). Emissions of NOx and the resulting acidic $HNO_3$, are also declining. In contrast, ammonia, the primary alkaline fine mode aerosol precursor, was either constant or increasing during this period (Pinder et al. 2007, 2008, Heald et al. 2012), owing to intensified agricultural activity and livestock farming from the demands of population growth. These trends have created the expectation that the aerosol has and will become increasingly less acidic (West et al. 1999, Pinder et al. 2007, 2008, Heald et al. 2012, Tsimpidi et al. 2007, Saylor et al. 2015), with ammonium sulfate being replaced, at least in part, by ammonium nitrate (West et al. 1999, Bauer et al. 2007, Bellouin et al. 2007, Li et al. 2014, Goto et al. 2016).

The concept of "nitrate substitution" of sulfate has largely been based on the notion that nitrate is volatile when the aerosol is acidic, and in turn aerosol is acidic when insufficient amounts of total ammonia (i.e., gas+aerosol) or dust non-volatile cations (NVCs) exist to neutralize aerosol sulfate. Based on this conceptual model, aerosol ionic molar ratios have largely been used as proxies of aerosol acidity (pH), so that when the aerosol ammonium to sulfate molar ratio approaches 2 (the composition of ammonium sulfate), aerosol is assumed "neutral" and only then nitrate aerosol can form (Fisher et al. 2011, Hennigan et al. 2015, Wang et al. 2016, Silvern et al. 2017). Modeling studies have corroborated this view, predicting that nitrate substitution may be prevalent in the future, including the Southeastern US (SE US) (Heald et al. 2014, Baeur et al. 2007, Bellouin et al. 2011, Li et al. 2014, Goto et al. 2016, Vayenas et al. 2005, Karydis et al. 2016). A more careful analysis however (Guo et al. 2015, Weber et al. 2016, Hennigan et al. 2015,

Guo et al. 2016) reveals that this conceptual model of aerosol acidity and conditions for nitrate
substitution fail; thermodynamic analysis of SE US aerosol observations instead show that fine
mode aerosol remains strongly acidic, despite a 70% reduction in sulfates, and more than sufficient
total ammonia to neutralize it. The strong acidity is maintained by the large difference in volatility
between sulfate and ammonia (Guo et al. 2015, Weber et al. 2016), so large changes in total
ammonia concentrations are required for a notable change in aerosol acidity, about one order of
magnitude increase in $NH_3$ concentration per unit increase in aerosol pH (Guo et al. 2015 &
2017c). However, ammonia gas deposits relatively rapidly, limiting its build up except in high
emissions regions. Throughout the decade, the levels of aerosol nitrate have remained relatively
constant throughout the US (Guo et al. 2015, Weber et al. 2016, Pye et al. 2009). The persistent
strong aerosol acidity in turn explains why nitrate aerosol has not considerably increased over the
last decades, and is unlikely to appear in the immediate future in the SE US. These findings
constitute a "paradox", as the same thermodynamic models (e.g., ISORROPIA-II Fountoukis &
Nenes 2007) used to demonstrate the aerosol tendency for strong acidity in the SE US (Guo et al.
2015, Weber et al. 2016) using ambient data, is also used in 3D modeling studies (Pye et al. 2009,
Heald et al. 2012) for the region that predict nitrate substitution as a possible aerosol response.
Reconciling the "nitrate substitution paradox" requires a careful examination of aerosol
thermodynamics and the conditions under which nitrate partitioning to the aerosol is favored.
Meskhidze et al. (2003) and later Guo et al. (2016) showed that for aerosol nitrate formation to
occur, aerosol pH needs to exceed a certain characteristic value (that depending on the temperature
and the amount of liquid water, ranges between a pH of 1.5 and 3; Guo et al., 2017). If aerosol pH
is therefore high enough (typically above a pH of 2.5 to 3), a behavior consistent with "nitrate
substitution" emerges, because any inorganic nitrate forming from NOx chemistry mostly resides
in the aerosol phase. When pH is low enough (typically below 1.5 to 2), nitrate remains exclusively
in the gas phase (as $HNO_3$), regardless of the amount produced, and "nitrate substitution" is not
observed. Between these "high" and "low" pH values, a "sensitivity window" emerges (of
typically 1-1.5 pH units), where partitioning shifts from nitrate being predominantly found as gas
to where it is mostly found as an aerosol. Therefore, if a model is for any reason biased in its
prediction of aerosol pH, it may be preconditioned towards nitrate prediction biases. The
sensitivity to pH biases is strongest when the aerosol lies in the pH "sensitivity window", which
is often the case for atmospheric aerosol (Guo et al. 2015, 2016 & 2017, Bougiatioti et al. 2016).
When below this "pH sensitivity window", aerosol nitrate is almost nonexistent and relatively
insensitive to emissions (and pH biases); when above the window, almost all nitrate resides in the
aerosol phase, and directly responds to NOx emission controls.

If aerosol were composed only of non-volatile sulfate and semi-volatile nitrate and

ammonium, prediction biases in pH could result only from errors in RH, and large errors (e.g.,
order of magnitude) of $NH_3$, NOx and $SO_2$ because pH is relatively insensitive to changes in these
aerosol precursors (Hennigan et al. 2015). Acidity however can also by modulated by other soluble
inorganic cations from seasalt and mineral dust, such as $K^+$, $Na^+$, $Ca^{+2}$ and $Mg^{+2}$. The low volatility
of these cations allows them to preferentially neutralize sulfates over $NH_3$, and, even in small
amounts elevate particle pH to levels that can promote the partitioning of nitrates to the aerosol
phase (Fountoukis & Nenes 2007, Guo et al. 2017a). NVCs tend to reside in the coarse mode
aerosol, with a fraction found in smaller particles, while sulfate tends to reside in the fine mode
(e.g., West et al. 1999, Vayenas et al. 2005, Guo et al. 2015); the degree to which NVCs can affect
fine mode pH therefore lies in the degree to which the two types of species mix across different
particle sizes. Potential interactions between inorganics-organics can also affect aerosol acidity.
However, recent studies driving thermodynamic models utilizing water associated with organics
find only minimal differences in pH predictions (Guo et al. 2015, Bougiatioti et al. 2016, Liu et
al. 2017, Pye et al., 2018, Song et al. 2018). In the presence of very high NVCs (for example in
sea-spray aerosol), where the aerosol has much higher pH, the pH can approach the pKa of organic
acids, leading to conditions where their dissociation can contribute to aerosol acidity (Laskin et
al. 2012).

Although aerosol models are evaluated in terms of their ability to predict the concentration

of aerosol species (including across size), no studies to date focus on their ability to predict aerosol
pH across size, even though it is known to potentially vary up to 6 units (Fang et al. 2017,
Bougiatioti et al. 2016, Li et al. 2017). Evaluation of models in this context is challenging, since
there is no established dataset of aerosol acidity - although that is rapidly changing, with pH
estimates derived from a combination of observations and models (e.g., Guo et al., 2015;
Bougiatioti et al., 2016; Guo et al., 2017; Liu et al., 2017; Song et al., 2018) -. Furthermore, given
that most of this pH variability occurs in the $PM_1$ to $PM_{2.5}$ range (Fang et al. 2017), it is quite
likely that model assumptions on how aerosol species interact within a mode (degree of internal
mixture), especially for particles in the 1-2.5 μm range, may lead to pH prediction biases that drive
model behavior.
This aim of this study is to address the underlying reasons for the "nitrate substitution"
paradox, and in the process, provide a conceptual framework for quantifying and understanding
the importance of aerosol pH biases. The guiding hypothesis of this work is that aerosol pH
prediction bias fundamentally changes predicted aerosol behavior and is the underlying cause of
the paradox. The approach is demonstrated with the Community Multiscale Air Quality (CMAQ)
model (Byun & Schere 2006) and is based on predictions of pH over the 2001-2011 period in the
Southeastern/Eastern US, being the region for which aerosol pH trends are constrained by
observations. The role of internally-mixed nonvolatile cations in $PM_{2.5}$ as a source of the pH bias
is then assessed.

## Methods


### Predicting aerosol pH and composition


CMAQ is a three-dimensional, Eulerian, atmospheric chemistry and transport model, that
simulates the processes atmospherically relevant compounds undergo, such as emission, diffusion,
chemical reactions and deposition (Byun & Schere 2006). CMAQ version 5.0.2 was used in this
study, and simulations were carried out using a 36-km horizontal resolution grid, with 13 vertical
layers, over the continental US (CONUS) for the entire years of 2001 & 2011. Meteorological
data were obtained offline from the Weather Research Forecasting (WRF) model. The same
meteorology was used between the two years to eliminate the effect of differences due to
temperature and relative humidity on pH predictions.. Model-ready emissions for 2011 were
obtained using the National Emissions Inventory 2011 inventory (NEI 2011) for the Carbon Bond
05 (CB05) chemical mechanism. To estimate the 2001 emissions, the 2011 emissions for $SO_2$,
NOx, $NH_3$, CO, VOCs and primary PM from anthropogenic sources were scaled on a per-species
basis using the Air Pollutant Emissions Trends Data (2017); emissions for other species were kept
constant. Specifically anthropogenic CO, $NO_x$, primary PM and $SO_2$ emissions were increased by
44%, 45%, 15% and 246% respectively,  while VOC and $NH_3$ emissions were reduced by 6% and
14% respectively. Emissions of biogenic species were calculated online using the Biogenic
Emission Inventory System (BEIS).
The aerosol thermodynamic model ISORROPIA-II (subversion 2.1 - dated 2008 –
Fountoukis & Nenes 2007) was used online in CMAQ to drive the semivolatile partitioning of
inorganic species, as well as offline to analyze the predicted $PM_{2.5}$ pH, nitrate partitioning
tendency and sensitivities thereof to nonvolatile cations. It should be noted that ISORROPIA and
CMAQ only account for the thermodynamic interactions between inorganic species and do not
treat organics. Offline calculations were conducted using the hourly gas and particle phase
concentrations output from CMAQ for the 2001 and 2011 simulations, which includes NVCs, and
using them as input to ISORROPIA-II (subversion 2.3 - dated 2012). The thermodynamic
calculations online and offline were carried out in forward mode, meaning that the temperature,
relative humidity, as well as all aerosol and gas phase concentrations were known and used as
input, while at the same time assuming that the aerosol is in a metastable state, where only one
aqueous phase is allowed to exist (Fountoukis & Nenes 2007). This assumption is not always
necessarily true, especially under conditions of low relative humidity (RH<30%) where the aerosol
can crystalize or exist in glassy, amorphous state (where in this case thermodynamic equilibrium
is not reached), observational data of liquid water content shows that it is most often a valid
assumption (Guo et al. 2015, Bougiatioti et al. 2016), and other studies suggest that the phase state
may not strongly affect predicted pH (Song et al., 2018).  We run the model under a variety of
conditions to determine the impact of NVCs from dust and sea salt (Ca, Mg, K, Na) on pH, its
seasonal variability, as well as the effect of pH and temperature on nitrate partitioning.

## Results and discussion

### Predicted Sulfate, ammonium & nitrate

For the main inorganic aerosol species ($SO_4^{-2}$, $NO_3^-$ and $NH_4^+$), CMAQ captures the

observed trends, as seen in the literature (Park et al. 2006, Hand et al. 2012, Blanchard et al. 2013a,
b, Kim et al. 2015, Saylor et al. 2015) over the CONUS over the course of the decade (Figure S1).
As expected, sulfate over the entire US drops significantly between 2001 and 2011 (~ 30%), with
major decreases in the Eastern US (~2 μg m$^{-3}$). Areas impacted the most by these reductions are
places of significant industrial activity or coal-fired electricity generating units (EGUs), such as
the Ohio River Valley, Baton Rouge in Louisiana and South Carolina. Ammonium levels only
experience small reductions which are a buffered response to the decrease in sulfate levels, and
minimal changes in emissions. Local reductions (~20%) in ammonia are seen over North Carolina
and Louisiana. Aerosol nitrate concentrations remain constant on average over the domain, with
small increases over the Eastern US. The highest levels of ammonium are observed in areas with
significant livestock, such as North Carolina and the Midwest; sulfate concentrations are the
highest around the Ohio River Valley, due to $SO_x$ emissions, and so is nitrate due to significant
$NO_x$ and ammonia emissions.

**Predicted Annual & seasonal pH**

Figure 1 depicts the annual average pH fields over the US for 2001 and 2011, calculated

using the annual average$PM_{2.5}$ concentrations, with the study domain of the Eastern US outlined.
Simulations show that there are noticeable differences between the two years, localized mainly in
desert regions along the US-Mexico border, Southern Texas and the Eastern US. The sulfate
reductions in the Eastern US, appear to have a major impact on model results, leading to significant
increases of aerosol pH in the area. For 2001, the average yearly pH for the Eastern US is 1.6,
consistent with recent literature and observations from the WINTER campaign (Guo et al. 2015
& 2016, Weber et al. 2016) (Figure 1a). For 2011, however, predicted pH increases to about 2.5 –
almost a unit higher (Figure 1b).

Seasonal pH trends are also positive over the Eastern US, with the summertime (Figure

S2f) experiencing stronger increases than in the winter (Figure S2c), being 0.5-1.5 for winter and
0.5-2 for summer. Much of the seasonal variability is driven by changes in temperature and relative
humidity; increased relative humidity (RH) leads to less acidic aerosol, since liquid water content
and pH are inversely related (Guo et al. 2015 & 2016), while increased temperatures promote low
RH and therefore more acidic aerosol. The desert areas of the Western US, Southern Texas,
Florida, SW Alabama and Mississippi are the most sensitive in the wintertime (Figure S2a, b),
while the Central US is mostly unaffected. During the summer, the entire Central US is much
more strongly impacted, while the wintertime sensitive areas exhibit only minor pH increases
(Figure S2d, e).
**Model evaluation of pH**

Model results for both simulation years were compared to thermodynamic analysis of

measurements from three urban sites in Atlanta, Georgia (Jefferson Street, JST; Georgia Tech,
GT; Atlanta Road-Side, RS) and two rural (Yorkville, Georgia - YRK; and Centerville, Alabama
- CTR) SEARCH network sites. Measurements for the urban sites and the YRK site, were taken
between May and December 2012 for the SCAPE study, while measurements from the CTR site
were for the SOAS campaign period (June 1[st] to July 15[th] 2013) (Guo et al. 2015, Xu et al. 2015).
The three urban sites are contained within the same CMAQ grid cell. All urban sites (Figure 2a,
b, c, d), exhibit an early morning/late night pH maximum, and an afternoon minimum throughout
the year (Guo et al. 2015). This a combination of two factors; RH being highest during the early
morning/late night, which increases water uptake and hence decreases acidity (Guo et al. 2015)
(Figure S3), and the presence of crustal elements in significant quantities during that time (Figure
S4). The model pH closely tracks the diurnal profile of predicted cations (Figure S4), indicating
that they have an important impact on predicted pH, which, however, is not seen in the
measurements (Figure 2), since they make up a much smaller percentage of observed PM$_{2.5}$.
Despite the presence of NVCs, the pH remains low for both simulation years but it tends to be
higher in 2011, because of sulfate levels that are approximately half of those in 2001 across all
sites, leading to the increased relative effect of NVCs (Weber et al. 2016). Removal of all NVCs
from the thermodynamic calculations (Figure S5),significantly reduces the pH differences
between 2001 and 2011 while removing some of the increased variability introduced by NVCs.
At the same time, a negative bias is introduced to the simulated pH, which is more prominent for
the urban sites even after the sulfate reductions.

The increase in pH is not proportional to the reduction in sulfate, since aerosol responds
non-linearly to such reductions, through the volatilization of ammonia (Weber et al. 2016).
Depending on location, sulfate reductions range from 38 to 55%, while the corresponding pH
increase is much lower, pointing to the fact that cations, although small in amount, tend to have a
disproportionately strong impact on acidity. For the SOAS campaign period (Figure 2g), pH is
underestimated especially for 2001. The biases follow the pattern of NVCs present, by being
negatively biased until noon and positively biased for the rest of the day (Figure 2 and Figure S4).
The bias is particularly evident in the early morning hours where NVC concentrations reach a
maximum (Figure S4).For the YRK site (Figure 2b, e), pH is overall underestimated during the
winter and overestimated during the summer. Similarly to the urban sites, the predicted RH agrees
well with the measurements (Figure S3), albeit with a positive afternoon bias during the summer.
The diurnal profile of pH closely tracks the one of cations, further suggesting they may be directly
related to the bias.
When evaluating the predicted pH trend for CTR, the model results exhibit a clear,
increasing trend of 0.6 pH units per decade (Figure 3). This trend is inconsistent with recent
thermodynamic analysis of observations suggesting a slight decrease in pH over the same time
period for the SE US (Guo et al. 2015 & 2016, Weber et al. 2016). If this bias in predicted pH
trend continues, it can have profound implications for future regulatory modeling, since the
increased pH can lead to elevated levels of model nitrate, reproducing nitrate substitution (Bauer
et al. 2007, Bellouin et al. 2011, Li et al. 2014, Goto et al. 2016). Possible reasons behind this pH
bias could be overestimated ammonia emissions, underestimated sulfate, or, the presence of NVCs
in $PM_{2.5}$. The first two possibilities are unlikely, given the agreement of predicted ammonium and
sulfate with previous studies (Park et al. 2006, Hand et al. 2012, Blanchard et al. 2013a, b, Kim et
al. 2015, Saylor et al. 2015), and, the relative insensitivity of pH to ammonia and sulfate (Weber
et al. 2016, Silvern et al. 2017). However, NVCs, if inappropriately distributed in $PM_{2.5}$, can exert
important biases on pH (Meskhidze et al. 2003, Karydis et al. 2016, Guo et al. 2017b). Indeed,
offline calculations of aerosol pH excluding the influence of NVCs mitigates most of the predicted
positive trend of 0.6 pH units per decade when all the aerosol species are considered (Figure 3),
while also reducing standard error. The remaining bias may arise from errors in model RH, given
that it controls water uptake and drives much of the diurnal variability in pH (Guo et al. 2015).
Usage of observed (instead of predicted) RH in the thermodynamic calculations, did not impact
the predicted pH more than 0.1 units (Figure S6). A more thorough evaluation of the remainder of
the pH bias, as well as the impact of NVCs when included in appropriate quantities, requires a far
more extensive analysis of the emissions profiles – especially regarding its diurnal variability -
and observational dataset than the one available for this study (Henneman et al. 2017, Guo et al.
2017c).
The pH bias becomes negative for most of the CMAQ Eastern US when removing all
NVCs from the calculations (Figure S5). This, combined with the considerable model skill for
sulfate, nitrate and ammonium when compared to literature (Henneman et al. 2017) implies that
pH biases are not related to errors in the major inorganic ions or biases in meteorological
parameters (humidity and temperature), but rather in the NVCs which are minor contributors to
$PM_{2.5}$, hence poorly constrained. For the SEARCH sites NVCs comprise 5 to 10% of the total
inorganic $PM_{2.5}$ (Guo et al. 2015), which is significantly less than what the model predicted values
that are a factor of 4 higher than the measurements. The most important result therefore is that
NVCs are a considerable source of pH prediction uncertainty when not accounted for correctly
(Supplementary material: The role of NVCs in $PM_{2.5}$ pH). It should be noted that for the
summertime at the CTR location, the ammonium and sulfate values are biased low in CMAQ by
a factor of 3 using the Weber et al. 2016 data. These biases however are consistent with literature
and typical of CTMs (Henneman et al 2017).
The SEARCH sites have been thoroughly studied in previous literature (Guo et al, 2015 &
2017a, Xu et al. 2015, Weber et al. 2016) and given the high concentrations of organic mass
observed throughout the year, they present an excellent case study for the potential impact of
organics on pH. Recent studies indicate that organic aerosol can have a secondary, but still
quantifiable impact on aerosol pH, especially when allowed to interact with inorganics (Pye et al.
2018). Most 3D models do not account for potential, non-ideal interactions between the two, in
addition to not including organics in thermodynamic calculations, which, if the above statement
is true, can lead to significant predictive pH errors.  To investigate the role of organics on pH we
used the E-AIM model (Wexler & Clegg 2002, Friese & Ebel 2010, Clegg et al. 1992)
(http://www.aim.env.uea.ac.uk/aim/aim.php) on our model results for the SEARCH sites, to
calculate partitioning with organics/inorganic interactions considered. We tested a variety of
organic compounds under different scenarios to determine the potential of organics to influence
pH (see SI: Organic acids and pH).
We find that addition of organic compounds to the model, did not have a significant impact
on acidity (≤2% pH deviation from the baseline value) compared to the baseline run, apart from
the cases where RH was higher than 80% and the mole fraction of organic acids in the aqueous
phase is greater than 25% (SI: Organic acids and pH). We conclude that the maximum impact of
organics on aerosol pH can likely result from the effects of liquid-liquid phase separation (Pye et
al. 2018), but of insufficient magnitude to sustain a positive aerosol pH trend as observed in our
basecase simulation.
**The impact of pH biases on nitrate partitioning and "sulfate-nitrate substitution"**
To understand the importance of pH biases on nitrate partitioning and the potential for
predicting a behavior consistent with "nitrate substitution", we express the CMAQ output in each
grid cell in terms of the nitrate partitioning ratio, $\varepsilon_{NO3} = \frac{[NO_3^-]}{[HNO_3]+[NO_3^-]}$. It can be shown that $\varepsilon_{NO3}$
follows a simple sigmoidal curve (Meskhidze et al. 2013, Guo et al. 2016), $\varepsilon_{NO3} = 1 - \frac{[H^+]}{[H^+]+L\cdot T\cdot \Psi}$,
where $L$ is the liquid water content, T is ambient temperature , $[H^+]$ is the concentration of $H^+$ in
the aerosol aqueous phase, and $\Psi = \frac{R \cdot [H_{NO_3}]}{1000 \cdot P_0}$ is a fitting parameter that depends on the universal
gas constant ($R$), the effective Henry's law constant for nitric acid in the aerosol aqueous phase
($H_{NO3}$) and the ambient pressure ($P_0$). Depending on the value of pH, nitrate partitioning in CMAQ
can either be insensitive ($\frac{\partial \varepsilon_{NO3}}{\partial pH} \sim 0$) or sensitive ($\frac{\partial \varepsilon_{NO3}}{\partial pH} \sim 0.5$) to pH biases, depending on the
month of the year considered (Figure 4). We generally find that nitrate partitioning is insensitive
($\frac{\partial \varepsilon_{NO3}}{\partial pH} \sim 0$) and heavily shifted to the gas phase ($\varepsilon_{NO3} \sim 0$) during the summer and spring (Figure
4), while it becomes quite sensitive to pH errors ($\frac{\partial \varepsilon_{NO3}}{\partial pH} \sim 0.5$) in the winter and fall. For the latter
case, this means that small pH perturbations in either direction can strongly affect the amount of
nitrate that partitions in the aerosol phase; if the weather is sufficiently cold and $NO_x$ emissions
and pH predictions are biased sufficiently high, $\varepsilon_{NO3} \sim 1$, meaning that all nitrates are partitioned
to the aerosol phase and the emergence of "nitrate substitution" behavior.

To exemplify the above, we determine the amount of excess nitrate from pH prediction

biases as follows. Perturbing the acidity by $\Delta pH$ from the monthly mean value along the $\varepsilon_{NO3}$
curves (Figure 4) gives the corresponding change in the partitioning ratio ($\Delta\varepsilon_{NO3}$). Multiplying
$\Delta\varepsilon_{NO3}$ with the total nitrate ($HNO_{3(g)}+NO_3$) predicted in CMAQ in each grid cell gives the total
nitrate response ($\Delta NO_3$) to $\Delta pH$. When applied to the Eastern US for $\Delta pH=+1$ (the average pH
impact of including NVCs in the $PM_{2.5}$ calculations over the entire Eastern US) gives the $\Delta NO_3$
distributions shown in Figure 5 for the winter (Figure 5a) and the summer (Figure 5b). The
predicted wintertime nitrate bias tends to be higher than in the summer, owing to the lower
temperatures and higher aerosol pH levels present (which shift $\varepsilon_{NO3}$ towards higher values; Figure
4) and the higher values of total available nitrate in the wintertime. The combination of both factors
(available nitrate and high pH) is necessary for appreciable quantities of nitrate to partition, but in
general the locations with a pH of between 0.5 and 1 are the most susceptible to positive pH biases,
since a unit increase places nitrate partitioning into the ascending part of the S-curve (Figure 4),
rapidly increasing the amount of aerosol nitrate that can form. During both seasons, areas rich in
total nitrate, and a pH between 0.5 and 1.5, such as the Ohio River Valley, New York, New Jersey
and South Louisiana (Figure 1, S1e, f), exhibit the largest increases in aerosol nitrate. Other
locations that have low pH, and low total nitrate such as West Virginia see minimal changes. A
notable exception is North Carolina which has a higher pH than the aforementioned locations -
mainly due to the high $NH_3$ emissions from livestock - which pushes the partitioning beyond the
sensitive regime, where increases in pH do not drive additional nitrate in the particle phase.

To investigate the potential of NVCs and sulfate reductions to induce nitrate substitution,

the sensitivity of the nitrate increase $\Delta NO_3$, to the corresponding sulfate reduction $\Delta SO_4$, was
quantified for the Eastern US, both when NVCs are included in the calculations and when they
were not (Figure 6). Over the decade, nitrate has seen increases in the Eastern US (Figure S11)
ranging from 0.5 to 2.5 $\mu g\ m^{-3}$, and NVCs can have a profound impact on how these increases are
distributed across the domain (Figure S11a, b). In the presence of NVCs (Figure 6a), there is a 1
$\mu g\ m^{-3}$ increase of nitrate for a sulfate reduction of the same value over the Eastern US. For this
case, substitution is predicted across the entire Eastern US, with only a few gridcells in South
Georgia, Mississippi and North Carolina exhibiting the opposite trend (nitrate reduction),
attributed to the formation of insoluble $CaSO_4$, which reduces the availability of aerosol water,
and in turn inhibits the formation of $NO_3$ with the co-condensation of $NH_3$. When NVCs are
removed (Figure 6b), the corresponding nitrate increase is much less (0-0.2 $\mu g\ m^{-3}$ per 1 $\mu g\ m^{-3}$ of
sulfate), especially in the Eastern US, while substitution is still predicted in the Northern parts of
the domain such as Ohio, Indiana and Michigan. The aforementioned areas, tend to have higher
seasonal pH values than the SE US (Figure 1), and the removal of NVCs reduces the pH to values
where nitrate partitioning is more sensitive to small pH perturbations (Figure 4), leading to a higher
predicted sensitivity to sulfate reductions. This analysis suggests that nitrate substitution is of a
much smaller magnitude than expected (West et al. 1999, Heald et al. 2012, Bauer et al. 2007,
Bellouin et al. 2011, Li et al. 2011, Goto et al. 2016, Vayenas et al. 2005, Karydis et al. 2016), and
heavily impacted by pH biases introduced from NVCs.

Given the importance of aerosol acidity for almost any aerosol-related process and impact,

it is imperative that aerosol studies evaluate acidity inferred from thermodynamic analysis of
ambient data as presented here. We demonstrate that in the case of nitrate substitution, the
distribution of nonvolatile cations over particle size can have a profound impact on model
behavior, and requires better constraints from emissions to observations (or at least appropriate
sensitivity studies, such as those carried out here, to unravel the potential impact of nonvolatile
cations). Understanding aerosol pH and the drivers thereof, is a powerful tool for evaluating model
performance that has never been used before. Usage of molar ratios, ion balances and other
conceptual models of aerosol acidity (Hennigan et al. 2015, Wang et al. 2016, Silvern et al. 2017)
provide limited insights in aerosol pH and should be strongly avoided to limit incorrect
conclusions.

## Acknowledgments

We acknowledge support from the Phillips 66 company, an EPA STAR grant and the European
Research Council Consolidator Grant 726165 – PyroTRACH. Authors declare no conflict of
interest. This work was funded, in part, by U.S. Environmental Protection Agency under grant
RD834799. Its contents are solely the responsibility of the grantee and do not necessarily represent
the official views of the US EPA. Further, the US EPA does not endorse the purchase of any
commercial products or services mentioned in the publication. Observational data were provided
by Atmospheric Research Associates.

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

**Figure 1** - Annual averaged $PM_{2.5}$ pH over CONUS for (a) 2001 and (b) 2011, calculated offline
using ISORROPIA, using the annual averaged CMAQ concentration fields. The white outline
specifies the Eastern US domain used for further analysis.

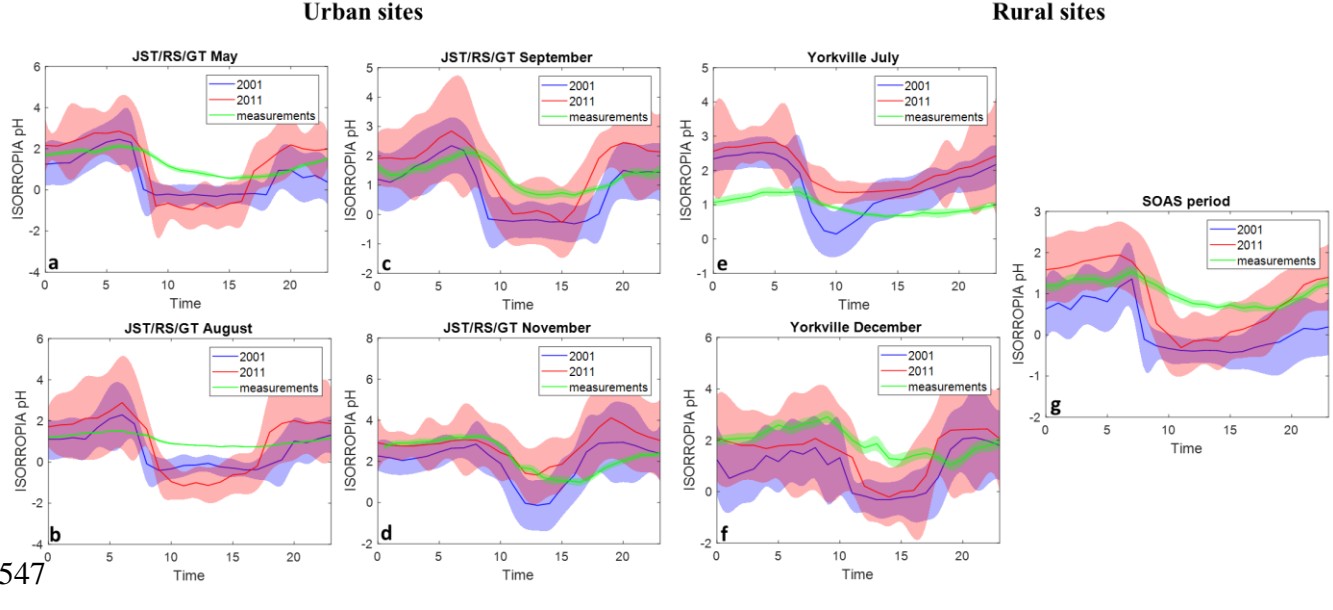



**Figure 2 -** pH diurnal profiles for May (a), August (b), September (c) and November (d) at
JST/RS/GT, July (e) and December (f) at YRK and for the SOAS campaign period (g). Blue and
red lines are the offline ISORROPIA simulated pH using CMAQ concentrations for 2001 and
2011 respectively, while the shaded areas are one model standard deviation. The green line
represents the pH calculated through the thermodynamic analysis of the measurements (found in
Guo et al., 2015) and the shaded area is standard standard error

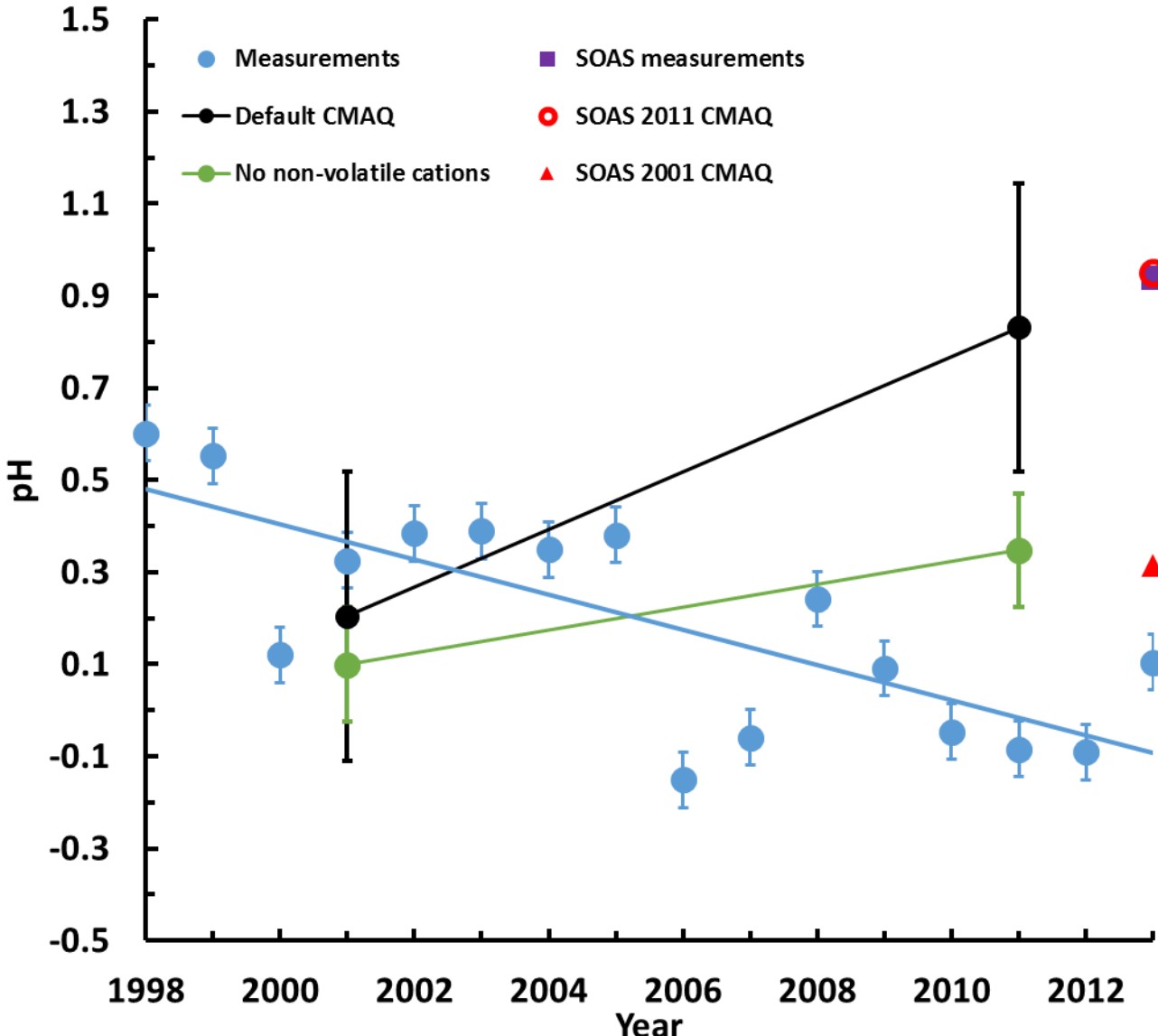


**Figure 3** – Decadal pH trends from the thermodynamic analysis of the measurements from Weber
et al. 2016 (blue line), default CMAQ (black line) and CMAQ results at the Centreville gridcell
without crustal elements (green line). Also shown, is the pH for the SOAS campaign, and for the
CMAQ predicted pH for June 1st-July 15th 2001 and 2011. CMAQ exhibits a clear positive trend,
with pH increasing throughout the decade, both due to sulfate reductions and the increasingly
important role of NVCs. Standard error is also plotted for all data points.


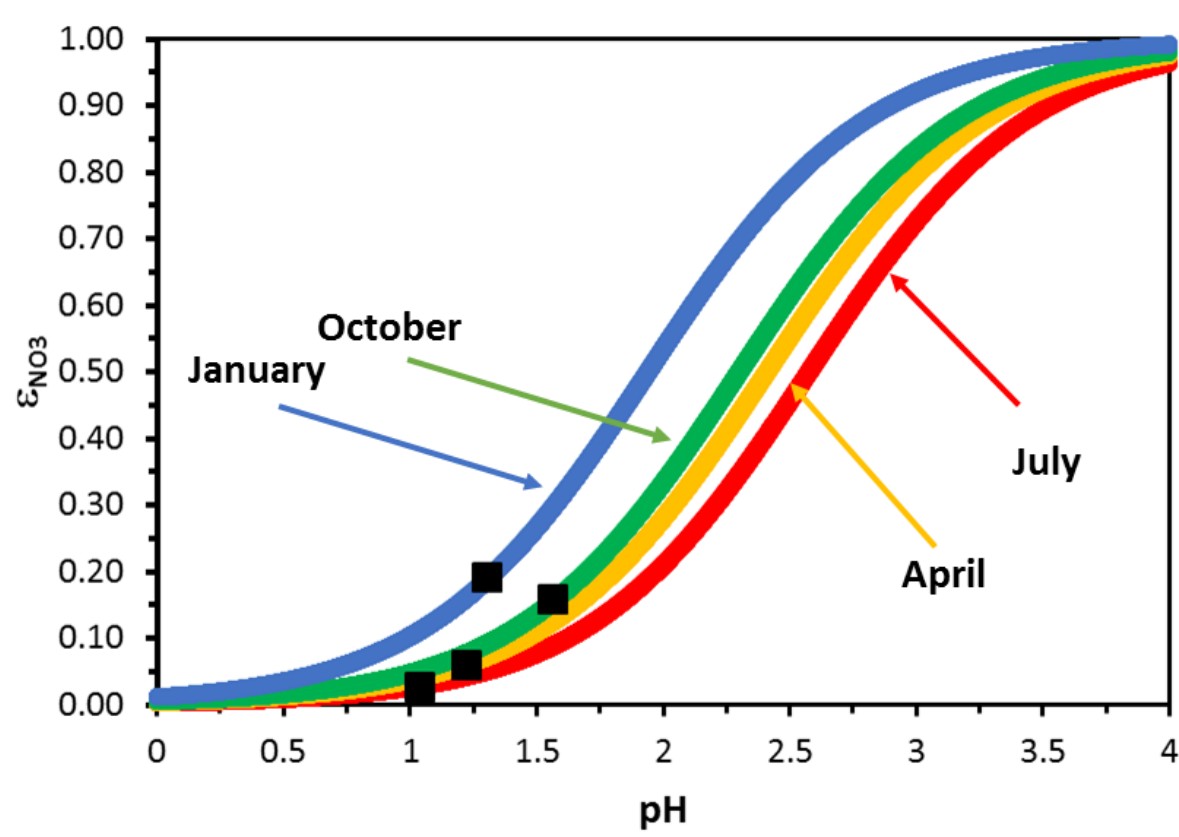


**Figure 4** - CMAQ-derived nitrate partitioning ratio for the E.US and select months of 2001. The
black squares denote the average pH values for each month. Note the insensitivity of nitrate
partitioning to pH biases in the summer for pH values of less than 1 ($\frac{\partial \varepsilon_{NO3}}{\partial pH} \sim 0$), which is not the case
for colder months.


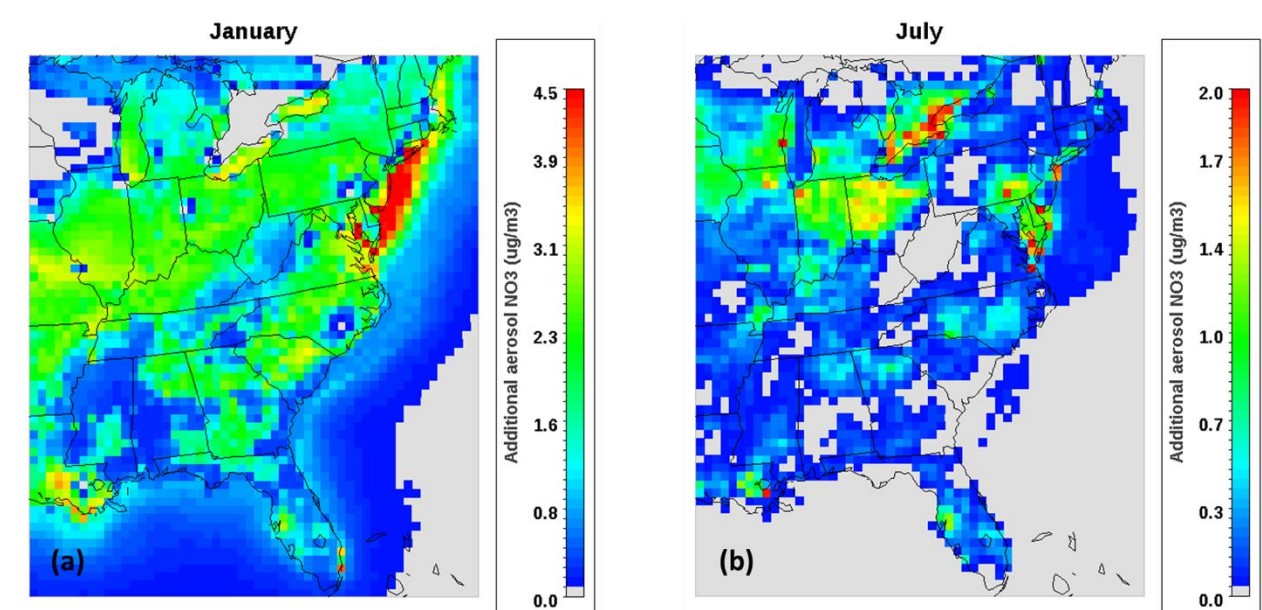


**Figure 5** - Increase in aerosol nitrate corresponding to a one-unit positive change in pH for a)
January and b) July. Emissions for 2011 are assumed, but to account for pH prediction biases from
NVCs, they are removed from the thermodynamic calculations. Plots are on different scales due
to the large differences in predicted nitrate increases.

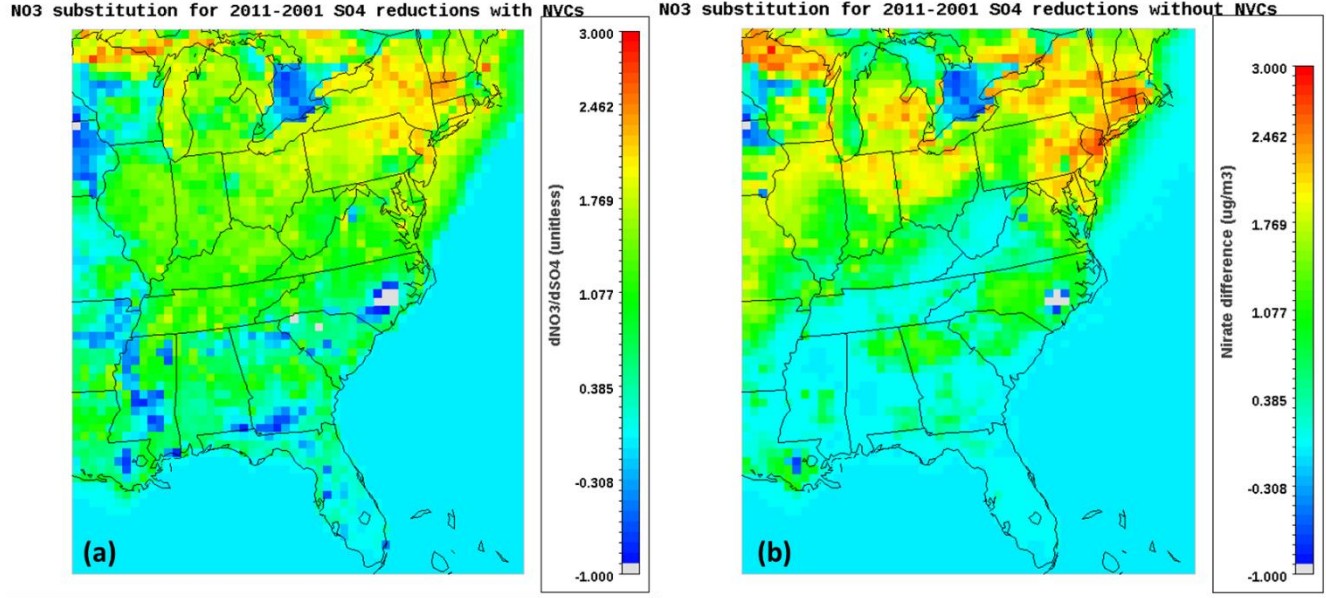

576

**Figure 6** – CMAQ predicted nitrate substitution ($\frac{NO_3^{2011}-NO_3^{2001}}{SO_4^{2001}-SO_4^{2011}}$) over the decade, when NVCs
are accounted for (a), and when they are removed from the thermodynamic calculations (b).