# Peer review of "Understanding nitrate formation in a world with less sulfate."

_Atmospheric Chemistry and Physics, 2018_

## Referee Comment (RC1) · Anonymous Referee #1 · 7 Jun 2018

This paper adds to an ongoing discussion of the importance of aerosol acidity on gas-particle partitioning, and how errors in modeled aerosol pH adversely affect model predictions. The subject is certainly of interest to readers of ACP.

However, the authors' arguments are not always stated clearly, particularly in the introduction. There are so many citations to prior literature throughout the paper that it is often unclear as to which aspects are new in the present manuscript. In some cases, studies are cited in support of a point that was not the conclusion (or even the subject) of the paper. The authors need to do a better job placing this work in the context of prior literature. In particular, one of the principal conclusions echoes the title of another paper currently under review for ACP by the same research group: Guo et al. (2017), The underappreciated role of non-volatile cations on aerosol ammonium-sulfate molar

ratios.

Specific Comments by line number

8 Suggest deleting "exacerbated by reductions in SO2 emissions." These reductions do not exacerbate the bias. The bias in pH leads to a bias in predicted response to SO2 emissions reductions.

10-11 This is the first of many sentences in this paper that overuse semicolons. More importantly, the authors do not have available direct observations of PM2.5 pH for either 2001 or 2011 and should not imply they do. Instead I recommend something like "modeled PM2.5 pH using 2001 emissions compare favorably with pH inferred from observed species concentrations. Using 2011 emissions, however, leads to simulated pH increases of one unit, which is inconsistent with observations from that year."

12-13 Instead of saying NVC are responsible for the (nonexistent) trend, clarify that overestimated NVC lead to this erroneous predicted increase in pH.

13-15 This sentence is unclear. Please rephrase or delete.

20-26 Consider breaking this sentence up. Also, it should not be necessary to cite two papers twice in the same sentence.

30 It is inappropriate to cite West et al. (1999) in support of an observed negative trend over 2001-2011 and beyond.

32 "ammonia ... is either constant or increasing." It is now mid-2018. Does this statement still hold, as implied by the use of present tense? Then please provide a more recent reference. Alternatively, if the authors intend to limit their statement to the 2001-2011 period, then use an appropriate tense to convey that meaning.

35 "will become increasingly neutralized." I believe one of the points the authors are making is that the conceptual model of "neutral" aerosol is inapt. If so, I suggest deleting this unhelpful phrase, so that the sentence reads "have created the expectation that

ammonium sulfate will be replaced, at least in part, by ammonium nitrate." Alternatively, please explain further what "neutralization" means in this context.

38-48 I realize that the authors are criticizing the adequacy of the conceptual model regarding molar ratios, but even so the arguments should be stated clearly. If, in this conceptual model, the molar ratio of ammonium to sulfate is what is salient, then how are NVCs at all relevant? In stating that modeling studies have corroborated this (mistaken) view, have these studies misapplied thermodynamic models? Or were those incorrect conclusions (that ammonium sulfate would be replaced by ammonium nitrate) based on the incomplete conceptual model rather than the full thermodymamic model? ISORROPIA itself uses the critical molar ratio in its calculations. Can the authors elaborate on when or to what extent this is appropriate?

112-120 The stated aim of this study is to address the underlying reasons for the nitrate subsition paradox. However, it seems that the resolution to this paradox has already been published in several papers by this research group, as summarized in lines 65-82, with further discussion in the subsequent paragraphs. Please clarify how this study represents an advance over that previous work. "The role of internally-mixed nonvolatile cations in PM2.5 as a source of the pH bias is then assessed." Again, to what extent is this new, and distinct from the Guo et al. (2017) manuscript under review?

135 Please provide a reference for the Air Pollutant Emissions Trends Data. If these are constant scaling factors, consider providing them in a table in the article Supplement.

138 Please clarify whether this is the same version of ISORROPIA as used in the release version of CMAQ v5.0.2.

143-148 Were these offline calculations computed separately for each of the three modes in CMAQ, or were the calculations done once by summing just the i and j modes? Given that CMAQ itself uses ISORROPIA, and HPLUS is one of the variables output by CMAQ, why was it necessary to run ISORROPIA again offline for each

grid cell and hour of each year? In presenting seasonal and annual average pH and nitrate partitioning ratios, are these calculated from hourly values?

160-171 The first sentence of this paragraph states that CMAQ captures the observed decreasing trends in SO4, NO3, and NH4. However, a few lines later the text states that ammonium levels "remain rather constant", and later in the same paragraph "aerosol nitrate concentrations remain relatively constant ... with small increases over the Eastern US." It is unnecessarily difficult to understand exactly what message the authors wish to convey.

161 Here and throughout the Results and Discussion section there are many references cited, but it is often not clear why they are being cited. In some instances, the authors mean that their current results are in agreement with results reported previously. However, the distinction between "Results" (of the current manuscript) and "Discussion" (including comparison with prior literature) should be stated explicitly. In this particular example, "CMAQ captures the observed downwards trends (six references cited) over the CONUS during the course of the decade," are those references supporting the assertion that inorganic species have been declining? Or that CMAQ generally represents species concentrations? This is especially confusing given the comment above, that ammonium and nitrate are largely unchanged between 2001 and 2011 in the current manuscript.

176 The methods section states that ISORROPIA is called offline using hourly CMAQ outputs, but the caption to Figure 1 indicates the calculation is performed using annual averaged CMAQ concentration fields (and presumably also annual average temperature and relative humidity). Given the strongly nonlinear dependence of aerosol partitioning on T and RH, I am skeptical of the value of a calculation based on annual average inputs to ISORROPIA.

183-187 "This trend suggests that pH will keep increasing... [four more references]." This discussion of the implications is repetitive and out of place here, especially since

the authors later argue that the modeled increase is in disagreement with WINTER observations and is erroneous, at least partially due to overestimated NVCs.

201-202 What is the meaning of citing Guo et al. (2015) and Xu et al. (2015) here? Those papers do not document the SEARCH data. Is the analysis performed here repeating work done in those papers?

205 The caption to Figure 2 states "Blue and red lines are the CMAQ predicted pH for 2001 and 2011 respectively," but the methods section states that ISORROPIA was called offline using CMAQ inputs, and the y-axis label indicates ISORROPIA. Which is it? Also, the caption appears to have been truncated.

207 Why is Guo et al. (2015) being cited here, along with Figure 2? Are the same data presented in Guo et al. (2015)? Or is the current result consistent with that previous study?

215 "due to the increased relative effect of NVCs (Weber et al. 2016)". Is the conclusion that NVCs are relatively more important made by Weber et al. (2016)? Again, please clarify what findings are new in the present manuscript.

215-217 Comparing Figure S5 to Figure 2, it is not at all obvious that the CMAQ values in S5 "better track" the observations by time of day than those in Figure 2. If this is an important point, it should be straightforward to substantiate it, such as via temporal correlations.

219 Of course the increase in pH is not proportional to the reduction in sulfate, since pH is logarithmic. The fact that aerosol responds non-linearly through volatilization of ammonia was stated previously.

224-225 It is strange to relate the pattern of the bias (difference between CMAQ predictions and SOAS measurements) to the pattern of the NVC concentrations. Moreover, I do not see the pattern referred to: the green line is between the red and blue lines up to about 8 in Fig. 2g, after which the model is negatively biased.

[Figure]

242-244 Foroutan et al. (2017) do not discuss pH at all.

293-294 If psi depends on the effective Henry's law constant for HNO3, which depends on H+, then psi is not a constant.

565 Figure 3 caption: are the Weber et al. (2016) mesasurements from the Centerville site? Do the CMAQ lines correspond to averages over the Eastern US domain or results at a single grid cell?

Technical Corrections by line number

22, 186 Surratt is misspelled.

90-93 Again, there is no need to cite the same three papers twice in the same sentence.

97 "find only minimal differences between predicted pH" is awkward. Perhaps "differences in predicted pH" or "differences in pH predictions."

130-131 "to eliminate potential biases of temperature and relative humidity on pH predictions." It would be clearer to state "to eliminate the effect of differences due to temperature and relative humidity on pH predictions."

149-153 This is a run-on sentence.

200 Clarify that these urban sites are in Atlanta, Georgia and the rural sites are also in Georgia.

202 sites is misspelled.

285 "out" should be "our"

287-289 The authors should be consistent as to whether this is "sulfate substitution" or "nitrate substitution."

303 Should this also be "nitrate substitution" rather than "nitrate partitioning"?

581 Caption is missing the word "change". The caption says "winter" and "summer"

but the figure titles say "January" and "July."

424-431 The same paper is listed twice, with differing author lists.

---

## Referee Comment (RC2) · Anonymous Referee #2 · 18 Jun 2018

The MS could be published without revision.

---

## Author Comment (AC1) · 2 Jul 2018

This paper adds to an ongoing discussion of the importance of aerosol acidity on gas- particle partitioning, and how errors in modeled aerosol pH adversely affect model predictions. The subject is certainly of interest to readers of ACP.

However, the authors' arguments are not always stated clearly, particularly in the introduction. There are so many citations to prior literature throughout the paper that it is often unclear as to which aspects are new in the present manuscript. In some cases, studies are cited in support of a point that was not the conclusion (or even the subject) of the paper. The authors need to do a better job placing this work in the context of prior literature. In particular, one of the principal conclusions echoes the title of another paper currently under review for ACP by the same research group: Guo et al. (2017), The underappreciated role of non-volatile cations on aerosol ammonium-sulfate molar ratios.

We thank the reviewer for his/her comments and we hope that, after the suggested changes to the paper, it has been scientifically strengthened while also exhibiting greater clarity.

Specific Comments by line number

8 Suggest deleting "exacerbated by reductions in SO2 emissions." These reductions do not exacerbate the bias. The bias in pH leads to a bias in predicted response to SO2 emissions reductions.

The reviewer is correct – we removed the statement.

10-11 This is the first of many sentences in this paper that overuse semicolons. More importantly, the authors do not have available direct observations of PM2.5 pH for either 2001 or 2011 and should not imply they do. Instead I recommend something like "modeled PM2.5 pH using 2001 emissions compare favorably with pH inferred from observed species concentrations. Using 2011 emissions, however, leads to simulated pH increases of one unit, which is inconsistent with observations from that year."

The sentence has now been revised according to the reviewer's comment.

12-13 Instead of saying NVC are responsible for the (nonexistent) trend, clarify that overestimated NVC lead to this erroneous predicted increase in pH.

The statement has been changed and now reads "Non-volatile cations ($K^+$, $Na^+$, $Ca^{+2}$, and $Mg^{+2}$) in the fine mode are found responsible for the erroneous predicted increase in aerosol pH of about 1 unit on average over the US"

13-15 This sentence is unclear. Please rephrase or delete.

Rephrased from "pH biases of 1 unit can induce a nitrate bias of 1-2 $\mu g\ m^{-3}$ which may further increase in future projections, reaffirming an otherwise incorrect expectation of "nitrate substitution""

to

"Such an increase can induce a nitrate bias of 1-2 $\mu g\ m^{-3}$ which may further increase in future projections, reaffirming an otherwise incorrect expectation of a significant "nitrate substitution""

20-26 Consider breaking this sentence up. Also, it should not be necessary to cite two papers twice in the same sentence.

The sentence has been revised to:

Aerosol acidity is a driver of many important atmospheric processes (Guo et al. 2015, Weber et al. 2016), catalyzing the conversion of isoprene oxidation products to form secondary organic aerosol (SOA) (Xu et al. 2015, Pye et al. 2013, Surratt et al. 2010, Eddingsaas et al. 2010), driving the semi-volatile partitioning of key aerosol species processes (Guo et al. 2015, Weber et al. 2016). High acidity can also lead to the solubilization of iron, copper and other trace metals in aerosol which may serve as nutrients for ecosystems (Meskhidze et al. 2003), but also prove toxic for humans (Ghio et al. 2012, Fang et al. 2017).

We feel that keeping the citations is appropriate in this case, since the literature for aerosol acidity is still developing in addition to each point referring to a different process. This way, the reader can more readily access the pertinent material if he/she so desires.

30 It is inappropriate to cite West et al. (1999) in support of an observed negative trend over 2001-2011 and beyond.

The reference has been removed, since as the reviewer pointed out, it is not appropriate here.

32 "ammonia ... is either constant or increasing." It is now mid-2018. Does this statement still hold, as implied by the use of present tense? Then please provide a more recent reference. Alternatively, if the authors intend to limit their statement to the 2001-2011 period, then use an appropriate tense to convey that meaning.

We updated the statement to reflect that it's intended for the 2001-2011 period.

35 "will become increasingly neutralized." I believe one of the points the authors are making is that the conceptual model of "neutral" aerosol is inapt. If so, I suggest deleting this unhelpful phrase, so that the sentence reads "have created the expectation that ammonium sulfate will be replaced, at least in part, by ammonium nitrate." Alternatively, please explain further what "neutralization" means in this context

Again, we thank the reviewer for identifying how the wording was less precise than desired. We replaced neutralized with "will become less acidic" for increased clarity, since this statement refers to an expected increase of aerosol pH.

38-48 I realize that the authors are criticizing the adequacy of the conceptual model regarding molar ratios, but even so the arguments should be stated clearly. If, in this conceptual model, the molar ratio of ammonium to sulfate is what is salient, then how are NVCs at all relevant?

NVCs are mentioned in line 40 with regards to their role in nitrate substitution and not molar ratios – they are not related to molar ratios as defined in the next sentences of the text.

In stating that modeling studies have corroborated this (mistaken) view, have these studies misapplied thermodynamic models? Or were those incorrect conclusions (that ammonium sulfate would be replaced

by ammonium nitrate) based on the incomplete conceptual model rather than the full thermodymamic model?

The crux of the paper is that models - even if the thermodynamics are rigorous – can be predisposed to incorrect predictions/conclusions if the simulated pH is sufficiently different from that of the ambient aerosol. For the case shown in this work, CMAQ's inherent pH bias is related to a high bias in the concentrations of NVCs in the fine mode. Therefore, the conclusions of the previous studies were consistent with the emissions used in the model, but because of the pH bias – not consistent with the ambient aerosol behavior.

ISORROPIA itself uses the critical molar ratio in its calculations. Can the authors elaborate on when or to what extent this is appropriate?

This is a great question. ISORROPIA does indeed use molar ratios to define the major salts that can form (and deliquesce) in the aerosol. In this sense, molar ratios are useful because they define the stoichiometry. After that step, the thermodynamic equilibrium relationships in the aqueous, solid and gas phases determine the relative concentrations of the ions in solution and, together with electroneutrality and mass conservation arguments, determine the aerosol pH.

112-120 The stated aim of this study is to address the underlying reasons for the nitrate subsitution paradox. However, it seems that the resolution to this paradox has already been published in several papers by this research group, as summarized in lines 65-82, with further discussion in the subsequent paragraphs. Please clarify how this study represents an advance over that previous work. "The role of internally-mixed nonvolatile cations in PM2.5 as a source of the pH bias is then assessed." Again, to what extent is this new, and distinct from the Guo et al. (2017) manuscript under review?

The work presented in Guo et al. (2017) investigates the role of NVCs in ambient measurements and how they can impact ammonia partitioning to address the Silvern et al. (2017) postulation that organics inhibit uptake of ammonia by aerosol, and that thermodynamic equilibrium for submicron aerosol simulations does not apply. The current manuscript, although focused on NVC, has a much different subject matter: it assumes thermodynamic equilibrium applies, but examines whether pH biases from having too much NVC in the fine mode biases the model response to nitrate aerosol. We show that although models, can provide predictions consistent with historically observed aerosol observations, will still incorrectly predict nitrate substitution for future sulfate reductions – and this is related to the simulated aerosol having too high of a pH (from too much NVC in the aerosol). The historical conditions were very high in sulfate/ammonium/nitrate such that a small bias in NVCs had relatively less impact on the simulations.

135 Please provide a reference for the Air Pollutant Emissions Trends Data. If these are constant scaling factors, consider providing them in a table in the article Supplement.

These data are directly provided by the EPA and the provided link serves as the reference – the constant scaling factors are now referenced in the manuscript (lines 151-153).

138 Please clarify whether this is the same version of ISORROPIA as used in the release version of CMAQ v5.0.2.

CMAQv5.0.2 uses ISORROPIA 2.1 while our study uses ISORROPIA 2.3 that includes the most recent bug-fixes. The changes between model versions do not affect the behavior in the model, and the conclusions of the study. This information has been added to the text.

143-148 Were these offline calculations computed separately for each of the three modes in CMAQ, or were the calculations done once by summing just the i and j modes? Given that CMAQ itself uses ISORROPIA, and HPLUS is one of the variables output by CMAQ, why was it necessary to run ISORROPIA again offline for each grid cell and hour of each year?

The offline calculations were done by summing the i and j modes in order to better approximate $PM_{2.5}$. As the reviewer correctly identifies, HPLUS is output by CMAQ, but only for the j mode, and by comparing just the j mode results with the measurements would render the comparison less accurate than the current approach. In addition, it would not be possible to calculate the impact of NVCs on aerosol pH or conduct sensitivity tests for nitrate substitution without offline tests.

In presenting seasonal and annual average pH and nitrate partitioning ratios, are these calculated from hourly values?

The seasonal values and nitrate partitioning ratios are calculated using hourly values. The annual average pH (Figure 1) is calculated using the annual average values of the CMAQ output.

160-171 The first sentence of this paragraph states that CMAQ captures the observed decreasing trends in SO4, NO3, and NH4. However, a few lines later the text states that ammonium levels "remain rather constant", and later in the same paragraph "aerosol nitrate concentrations remain relatively constant ... with small increases over the Eastern US." It is unnecessarily difficult to understand exactly what message the authors wish to convey.

We modified the paragraph for clarity, since it was indeed difficult to convey the appropriate message:

"For the main inorganic aerosol species ($SO4^{2-}$, $NO3^{-}$ and $NH4^{+}$), CMAQ captures the observed trends, as seen in the literature (Park et al. 2006, Hand et al. 2012, Blanchard et al. 2013a, b, Kim et al. 2015, Saylor et al. 2015) over the CONUS over the course of the decade (Figure S1). As expected, sulfate over the entire US drops significantly between 2001 and 2011 (~ 30%), with major decreases in the Eastern US (~2 µg m$^{-3}$). Areas impacted the most by these reductions are places of significant industrial activity or coal-fired electricity generating units (EGUs), such as the Ohio River Valley, Baton Rouge in Louisiana and South Carolina. Ammonium levels only experience small reductions which are a buffered response to the decrease in sulfate levels, and minimal changes in emissions. Local reductions (~20%) in ammonia are seen over North Carolina and Louisiana. Aerosol nitrate concentrations remain constant on average over the domain, with small increases over the Eastern US. The highest levels of ammonium are observed in areas with significant livestock, such as North Carolina and the Midwest; sulfate concentrations are the

highest around the Ohio River Valley, due to SO$_x$ emissions, and so is nitrate due to significant NOx and ammonia emissions."

161 Here and throughout the Results and Discussion section there are many references cited, but it is often not clear why they are being cited. In some instances, the authors mean that their current results are in agreement with results reported previously. However, the distinction between "Results" (of the current manuscript) and "Discussion" (including comparison with prior literature) should be stated explicitly. In this particular example, "CMAQ captures the observed downwards trends (six references cited) over the CONUS during the course of the decade," are those references supporting the assertion that inorganic species have been declining? Or that CMAQ generally represents species concentrations? This is especially confusing given the comment above, that ammonium and nitrate are largely unchanged between 2001 and 2011 in the current manuscript.

This is an excellent point. The references were added as an evaluation step for our results, in order to show that the model behavior of inorganic species is consistent with what has been observed. The statement has been clarified (see previous comment).

176 The methods section states that ISORROPIA is called offline using hourly CMAQ outputs, but the caption to Figure 1 indicates the calculation is performed using annual averaged CMAQ concentration fields (and presumably also annual average temperature and relative humidity). Given the strongly nonlinear dependence of aerosol partitioning on T and RH, I am skeptical of the value of a calculation based on annual average inputs to ISORROPIA.

All the plots in the paper are generated using hourly data or are the result from averaging that data (e.g. Figure 5 & 6 the fields resulting from averaging the hourly values of additional nitrate and nitrate substitution respectively), apart from Figure 1. The reviewer is correct in pointing out that there is a nonlinear dependence of partitioning on T and RH, and that is why we used the same meteorology between both years, so that the only thing that changes is the emissions of key species. Therefore Figure 1 still presents a useful qualitative plot in order to show the average increase of aerosol pH over the decade. We now mention this in this section to avoid confusion.

183-187 "This trend suggests that pH will keep increasing... [four more references]." This discussion of the implications is repetitive and out of place here, especially since the authors later argue that the modeled increase is in disagreement with WINTER observations and is erroneous, at least partially due to overestimated NVCs

This part of the text has been removed to avoid repetition.

201-202 What is the meaning of citing Guo et al. (2015) and Xu et al. (2015) here? Those papers do not document the SEARCH data. Is the analysis performed here repeating work done in those papers?

We used the data and analysis of these studies in order to compare our results and therefore it is appropriate that we reference them at this point.

205 The caption to Figure 2 states "Blue and red lines are the CMAQ predicted pH for 2001 and 2011 respectively," but the methods section states that ISORROPIA was called offline using CMAQ inputs, and the y-axis label indicates ISORROPIA. Which is it? Also, the caption appears to have been truncated

All the pH values provided in Figure 2 come from offline ISORROPIA runs. We now clarify that the CMAQ predicted pH refers to the pH calculated using CMAQ outputs and ISORROPIA. The caption has also been fixed.

207 Why is Guo et al. (2015) being cited here, along with Figure 2? Are the same data presented in Guo et al. (2015)? Or is the current result consistent with that previous study?

The green trendlines in Figure 2 are the same data from Guo et al. 2015. This is now clarified in the figure caption.

215 "due to the increased relative effect of NVCs (Weber et al. 2016)". Is the conclusion that NVCs are relatively more important made by Weber et al. (2016)? Again, please clarify what findings are new in the present manuscript.

This is a statement made to explain the increase between the two simulation years. While this conclusion was made in Weber et al. (2016), it was not observed behavior in their dataset and it did not pertain to model results as in our study. In that light it constitutes a new finding.

215-217 Comparing Figure S5 to Figure 2, it is not at all obvious that the CMAQ values in S5 "better track" the observations by time of day than those in Figure 2. If this is an important point, it should be straightforward to substantiate it, such as via temporal correlations.

The reviewer is correct – the intent was to show that removal of NVCs from the calculations reduces the pH differences between years, while at the same time removing some of the additional variability introduced by NVCs. The statement has been revised to "Removal of all NVCs from the thermodynamic calculations (Figure S5), significantly reduces the pH differences between the 2001 and 2011 while removing some of the increased variability introduced by NVCs."

219 Of course the increase in pH is not proportional to the reduction in sulfate, since pH is logarithmic. The fact that aerosol responds non-linearly through volatilization of ammonia was stated previously.

The sentence that follows the sentence quantifies the reduction in sulfate versus the reduction in pH in the model results; we found this important to mention.

224-225 It is strange to relate the pattern of the bias (difference between CMAQ predictions and SOAS measurements) to the pattern of the NVC concentrations. Moreover, I do not see the pattern referred to: the green line is between the red and blue lines up to about 8 in Fig. 2g, after which the model is negatively biased.

The positive bias in the early morning hours for SOAS coincides with the increased presence of NVCs. We rewrote the statement to clarify "The biases follow the pattern of NVCs present, by being negatively

biased until noon and positively biased for the rest of the day (Figure 2 and Figure S4). The bias is particularly evident in the early morning hours where NVC concentrations reach a maximum (Figure S4)."

242-244 Foroutan et al. (2017) do not discuss pH at all.

The reference has been removed.

293-294 If psi depends on the effective Henry's law constant for HNO3, which depends on H+, then psi is not a constant.

Indeed! We now clarify that for the needs of the paper we use it as a fitting parameter.

565 Figure 3 caption: are the Weber et al. (2016) mesasurements from the Centerville site? Do the CMAQ lines correspond to averages over the Eastern US domain or results at a single grid cell?

Yes the measurements are from the Centreville site. The CMAQ lines correspond to results from the gridcell where Centreville is located. We now clarify this in the caption.

Technical Corrections by line number

22, 186 Surratt is misspelled.

We apologize for the typo – it has been corrected.

90-93 Again, there is no need to cite the same three papers twice in the same sentence.

Indeed! We have reduced the references.

97 "find only minimal differences between predicted pH" is awkward. Perhaps "differences in predicted pH" or "differences in pH predictions."

Revised to "differences in pH predictions".

130-131 "to eliminate potential biases of temperature and relative humidity on pH predictions." It would be clearer to state "to eliminate the effect of differences due to temperature and relative humidity on pH predictions."

Updated statement to the one suggested by the reviewer.

149-153 This is a run-on sentence.

The sentence has been revised to "The thermodynamic calculations online and offline were carried out in forward mode, meaning that the temperature, relative humidity, as well as all aerosol and gas phase concentrations were known and used as input, while at the same time assuming that the aerosol is in a metastable state, where only one aqueous phase is allowed to exist (Fountoukis & Nenes 2007)."

200 Clarify that these urban sites are in Atlanta, Georgia and the rural sites are also in Georgia.

Location added for all the sites.

202 sites is misspelled.

Changed "cites" to "sites".

285 "out" should be "our"

Corrected.

287-289 The authors should be consistent as to whether this is "sulfate substitution" or "nitrate substitution."

The title of the section has been changed for consistency to "The impact of pH biases on nitrate partitioning and "sulfate nitrate substitution"".

303 Should this also be "nitrate substitution" rather than "nitrate partitioning"?

Changed from "nitrate partitioning" to "nitrate substitution".

581 Caption is missing the word "change". The caption says "winter" and "summer but the figure titles say "January" and "July."

Caption has been updated and now also reads January and July (the mid-month of the season were picked for this plot).

424-431 The same paper is listed twice, with differing author lists

Thank you for pointing this out! The duplicate has now been removed.

---

## Author Comment (AC2) · 2 Jul 2018

The MS could be published without revision

We would like to sincerely thank the reviewer for his/her recommendation to publish the paper. All the changes suggested by the first reviewer have been implement and it is our hope that the quality of the manuscript has further been improved.